# Dual control of pcdh8l/PCNS expression and function in *Xenopus laevis* neural crest cells by adam13/33 via the transcription factors tfap2α and arid3a

Vikram Khedgikar[1], Genevieve Abbruzzese[2], Ketan Mathavan[1,3], Hannah Szydlo[1], Helene Cousin[1], Dominique Alfandari[1,3]*

[1]Department of Veterinary and Animal Sciences, University of Massachusetts, Amherst, United States; [2]David H. Koch Institute for Integrative Cancer Research, Massachusetts Institute of Technology, Cambridge, United States; [3]Molecular and Cellular Biology graduate program, University of Massachusetts, Amherst, United States

**Abstract** Adam13/33 is a cell surface metalloprotease critical for cranial neural crest (CNC) cell migration. It can cleave multiple substrates including itself, fibronectin, ephrinB, cadherin-11, pcdh8 and pcdh8l (this work). Cleavage of cadherin-11 produces an extracellular fragment that promotes CNC migration. In addition, the adam13 cytoplasmic domain is cleaved by gamma secretase, translocates into the nucleus and regulates multiple genes. Here, we show that adam13 interacts with the arid3a/dril1/Bright transcription factor. This interaction promotes a proteolytic cleavage of arid3a and its translocation to the nucleus where it regulates another transcription factor: tfap2α. Tfap2α in turn activates multiple genes including the protocadherin pcdh8l (PCNS). The proteolytic activity of adam13 is critical for the release of arid3a from the plasma membrane while the cytoplasmic domain appears critical for the cleavage of arid3a. In addition to this transcriptional control of pcdh8l, adam13 cleaves pcdh8l generating an extracellular fragment that also regulates cell migration.

DOI: https://doi.org/10.7554/eLife.26898.001

*For correspondence:
alfandar@vasci.umass.edu

**Competing interests:** The authors declare that no competing interests exist.

## Introduction

Cranial neural crest (CNC) cells are a transient population of natural stem cells that are induced at the border of the neural and non-neural ectoderm by a combination of signals including BMP, Wnt and FGF (*Dorsky et al., 1998*; *Monsoro-Burq et al., 2003*; *Tribulo et al., 2003*). These signals mediate the expression of key transcription factors that define placodal and neural crest cells (*Groves and LaBonne, 2014*; *Milet and Monsoro-Burq, 2012*). Among those the transcription factor tfap2α is one of the earliest and most conserved factors that controls the fate of neural border cells and later neural crest cells (*de Croze et al., 2011*). While loss of tfap2α in all vertebrates tested leads to craniofacial defects, mutations in the tfap2α gene in humans are associated with the Branchiooculofacial Syndrome highlighting the importance of this gene for craniofacial development and neural crest cell biology throughout evolution (*Hoffman et al., 2007*; *Luo et al., 2003*; *Martinelli et al., 2011*; *Meshcheryakova et al., 2015*; *Van Otterloo et al., 2012*).

Once induced, CNC cells migrate toward the ventral side of the embryo to contribute to most of the craniofacial structures. Many proteins have been shown to contribute to CNC migration, including multiple cell adhesion molecules such as integrins, cadherins and protocadherins, as well as cell surface metalloproteases that can effectively modulate these adhesion molecules (*Alfandari et al.,*

2010). In particular, while cadherin-11 is required at the surface of CNC, an excess of the protein prevents migration (*Borchers et al., 2001*). The cytoplasmic domain of cadherin-11 binds to the GEF trio and promotes filopodia formation in the migrating CNC (*Kashef et al., 2009*). In addition, cadherin-11 also localizes to nascent focal adhesion and interacts with the proteoglycan syndecan-4 via its transmembrane domain (*Langhe et al., 2016*). Lastly, the extracellular domain of cadherin-11 is cleaved by the metalloprotease adam13 producing an extracellular fragment that promotes cell migration (*McCusker et al., 2009*). This fragment can compete with the integral cadherin-11 to decrease contact inhibition of locomotion by a homophilic-dependent binding site, but this site is not required for its ability to promote CNC migration (*Abbruzzese et al., 2016*).

Other cadherin superfamily members are cleaved by ADAM metalloproteases. In Xenopus, the protocadherin pcdh8l is essential for CNC migration both in vivo and in vitro suggesting that it affects the basic cell locomotion machinery (*Rangarajan et al., 2006*). Interestingly, the protocadherin pcdh8 (PAPC), which can functionally replace pcdh8l in the neural crest cells (*Schneider et al., 2014*) is a substrate for adam13 (*Abbruzzese et al., 2014*). In chick, cleavage of cadherin-6 by adam10 and adam19 is also critical for proper migration of the neural crest cells (*Schiffmacher et al., 2014*). Thus, controlled expression of cadherin and protocadherin combined with proteolytic processing of these cell adhesion molecules appears to play a critical role in neural crest cell migration.

In addition to its proteolytic cleavage of cadherin-11, adam13/33 has been shown to play critical roles in CNC induction in *Xenopus tropicalis* by cleaving ephrinB, reducing its inhibitory activity upon the Wnt signaling pathway and allowing for a robust expression of snai2 (*Wei et al., 2010*). In contrast, knock down of adam13 in *Xenopus laevis* has no effect on snai2 expression but affects CNC migration in vivo (*Cousin et al., 2011*; *Alfandari et al., 2001*; *Cousin et al., 2012*). While the proteolytic role of adam13 is clearly important, the role of its cytoplasmic domain is also critical. We have shown that the cytoplasmic domain of adam13 is cleaved by gamma secretase and translocates into the nucleus where it regulates the expression of thousands of genes. The cleavage by gamma secretase requires a self-proteolytic cleavage by adam13 in its own cysteine rich domain and is therefore dependent on the adam13 proteolytic activity. In the absence of the cytoplasmic domain, adam13 cannot support CNC migration, but addition of a soluble form of its cytoplasmic domain fused to GFP, which partition exclusively in the nucleus, rescues cell migration. Similarly, addition of a single leucine in the cytoplasmic domain of adam13 that creates a nuclear export signal prevents adam13 ability to rescue CNC migration. This ability to regulate cell migration and gene expression is shared by the cytoplasmic domains of adam13 orthologues from *C-elegans* to marsupial, as well as the cytoplasmic domains of adam19 from frog and mouse, demonstrating the evolutionary conservation of this role. In contrast, the cytoplasmic domains of adam9 and adam10, which also translocate into the nucleus, cannot compensate for the loss of the adam13 cytoplasmic domain, revealing the specificity of this activity (*Cousin et al., 2011*).

Because, the cytoplasmic domain of adam13 does not possess any sequence similarity with transcription factors. We have investigated how it can modulate the expression of so many target genes. Here, we show that adam13 regulates the expression of the transcription factor tfap2α, which in turn regulates the protocadherin pcdh8l, a target of adam13 critical for CNC migration. We further show that adam13 physically interacts with the transcription factor arid3a/dril1/BRIGHT. This transcription factor is required for adam13's ability to induce tfap2α. In human Hek293T cells, adam13 stimulates the production of a shorter fragment of arid3a at the plasma membrane, which translocates to the nucleus. The production of the shorter fragment depends on the adam13 cytoplasmic domain, while the translocation from the membrane depends on the adam13 proteolytic activity. We propose that adam13 may interact with arid3a at the plasma membrane providing a pool of transcription factor that can be released upon cleavage of the adam13 cytoplasmic domain. This function is likely conserved and could contribute to many of the biological processes that are regulated by ADAM proteins (*Alfandari et al., 2009*; *Dreymueller et al., 2012*; *Giebeler and Zigrino, 2016*; *Lu et al., 2008*; *Pollheimer et al., 2014*; *Reiss and Saftig, 2009*).

# Results

## Adam13 cleaves pcdh8l

We have previously shown that adam13 cleaves the protocadherin pcdh8 (*Abbruzzese et al., 2014*). Given the similarity in sequence between pcdh8 and pcdh8l and the critical role that pcdh8l plays during CNC migration, we hypothesized that adam13 may also cleave pcdh8l. To test this hypothesis, we co-transfected an N-terminal flag-tagged-pcdh8l (*Figure 1A*) along with either adam13 (A13) or a proteolytically inactive mutant form adam13E/A (A13E/A) in human Hek293T cells. Western blot analysis of conditioned media from transfected cells shows the presence of the extracellular fragment of pcdh8l only when co-transfected with adam13 (*Figure 1B*). The approximate size of the

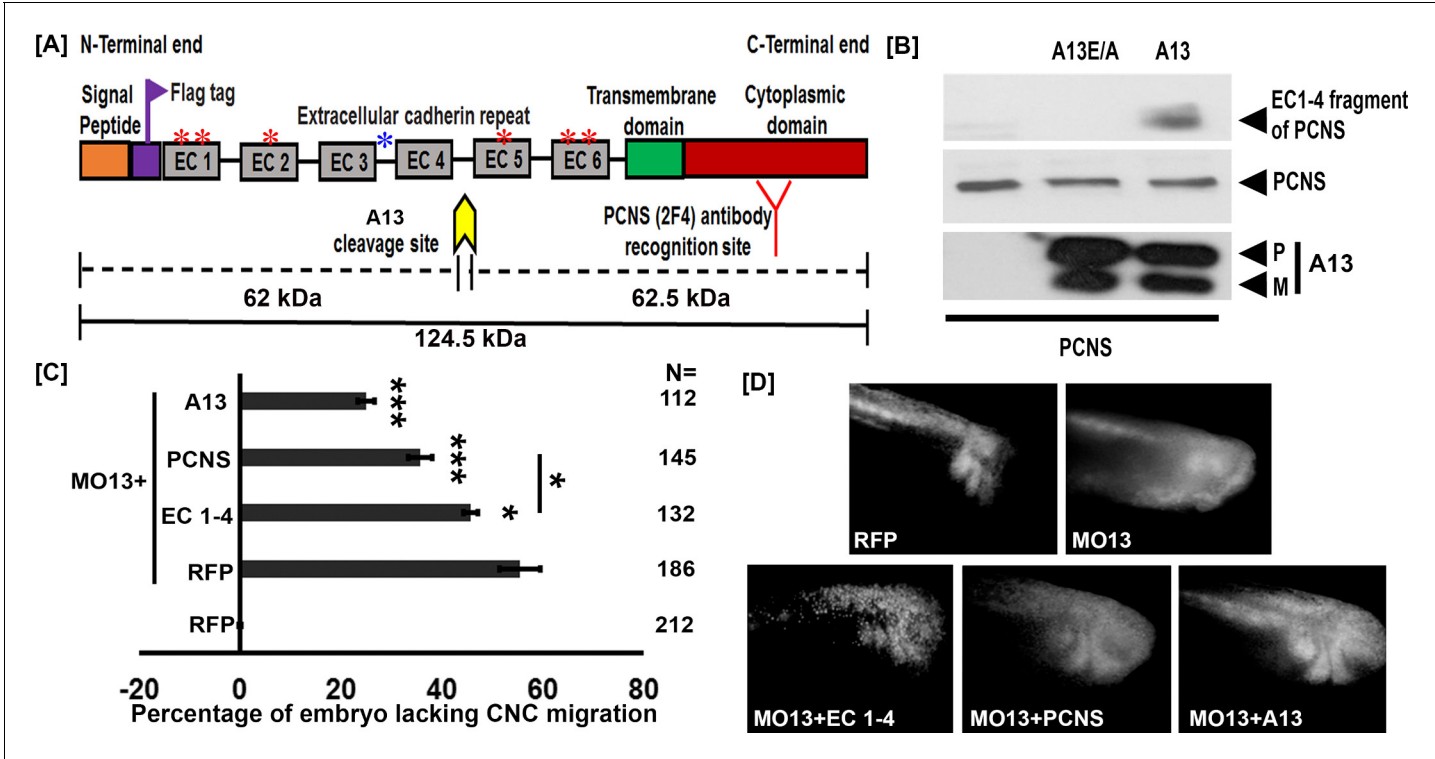

**Figure 1.** adam13 cleaves pcdh8l. (A) Schematic representation of full length pcdh8l. A Flag tag (DYKDDDDK) was introduced just before the first cadherin repeat (EC1). The yellow arrow indicates the predicted cleavage site of adam13 based on the molecular weight of N- and C- terminal fragments. Red and blue asterisk indicate N and O glycosylation site, respectively. The monoclonal antibody 2F4 was produced against the cytoplasmic domain (Red). (B) Western blot from transfected Hek293T cells. Glycoproteins from the conditioned media were purified on concanavalin-A agarose. The cleavage fragment was detected using the N-terminus Flag tag with the mAb M2. Total pcdh8l and adam13 were detected using mAb 2F4 and mAb 4A7, respectively. The Pro (P) and Metalloprotease (M) active forms of adam13 are indicated. (C) Histogram representing the percentage of embryos lacking fluorescent neural crest cell in the migration pathway. (D) Photographs of representative embryos with or without fluorescent migrating neural crest cells. Embryos were injected at the one-cell stage with a morpholino targeting adam13 (10 ng MO13). Messenger RNA for RFP alone or combined with the different constructs was injected at the 8-cell-stage in a dorsal animal blastomere. Each injection was compared to RFP injected in control embryos in which RFP positive cranial neural crest cells have successfully migrated into the branchial and hyoid arches (0% inhibition). N = number of embryos scored from three or more independent experiments. Error bars represent standard error to the mean. One-way ANOVA was performed to determine statistical significance. Statistically significant at *p<0.05, ***p<0.005.
DOI: https://doi.org/10.7554/eLife.26898.002

The following source data and figure supplements are available for figure 1:

**Source data 1.** Source data for *Figure 1*.
DOI: https://doi.org/10.7554/eLife.26898.005

**Figure supplement 1.** Characterization of mAb 2F4 to pcdh8l.
DOI: https://doi.org/10.7554/eLife.26898.003

**Figure supplement 2.** adam13 overexpression increases pcdh8l fragments.
DOI: https://doi.org/10.7554/eLife.26898.004

fragment (>60 kDa) suggests that it includes the cadherin repeats 1 to 4 of pcdh8l (EC1-4). This was confirmed by generating a truncated form of pcdh8l including these domains, which migrates at the same position as the cleaved fragment (data not shown). No fragment was observed when pcdh8l was co-transfected with A13E/A suggesting that the proteolytic activity of adam13 is responsible for the cleavage of pcdh8l. To confirm these findings in vivo, we generated a monoclonal antibody to the cytoplasmic domain of pcdh8l (mAb 2F4). This antibody recognizes a protein of the appropriate size (120 kDa), expressed at the appropriate stage (stage 20), localized to the CNC in whole mount immunostaining and to the plasma membrane of CNC cells in vitro. We further confirmed that it recognizes pcdh8l but not pcdh8 transfected in Hek293T cells (*Figure 1—figure supplement 1*). At stage 20, the main protein recognized is the full length at 120 kDa, but two minor shorter fragments can be seen at ≈60 (1) and ≈40 kDa (2), respectively (*Figure 1—figure supplement 2*). These fragments increased in embryos injected with the adam13 mRNA suggesting that adam13 can also cleave pcdh8l in vivo.

## The pcdh8l fragment can partially compensate for the loss of adam13

Adam13 cleaves cadherin-11 to generate a fragment that promotes CNC migration (*McCusker et al., 2009*; *Abbruzzese et al., 2016*; *Cousin et al., 2011*). To test if the extracellular fragment of pcdh8l also regulates CNC migration we attempted to rescue the CNC migration defect induced by the loss of adam13 by expressing various forms of pcdh8l. To that effect, we injected an antisense morpholino oligonucleotide known to block adam13 translation (MO13) (*McCusker et al., 2009*) at the one-cell stage and the various mRNA at the 8-cell-stage in a dorsal animal blastomere (*Moody, 1987*). A lineage tracer (RFP mRNA) was co-injected in all cases to follow the migration of the CNC as previously described (*McCusker et al., 2009*; *Abbruzzese et al., 2016*; *Abbruzzese et al., 2014*; *Cousin et al., 2011*; *Cousin et al., 2012*). When MO13 is injected alone 57% of the embryos show no fluorescent CNC in the migration pathways (*Figure 1C–D*). Injection of a MO13-resistant adam13 mRNA significantly rescued CNC migration (26% inhibition). Injection of the pcdh8l extracellular fragment (EC1-4) mRNA rescued CNC migration significantly (47% inhibition), but less efficiently than the adam13 mRNA. Surprisingly, we found that loss of adam13 could be significantly rescued by the full length pcdh8l mRNA even more efficiently (36% inhibition) than with EC1-4, a phenomenon that was not observed for cadherin-11 (*McCusker et al., 2009*). These results suggest that the loss of adam13 does not simply prevent the cleavage of pcdh8l but could also affect its expression, localization or function.

## Adam13 regulates pcdh8l expression

We first tested if the loss of adam13 affects pcdh8l expression. For this, MO13 (10 ng) was injected at the one-cell-stage, embryos were raised until stage 20 when pcdh8l expression is maximal (*Rangarajan et al., 2006*), at which time mRNA and protein were extracted for analysis. In these conditions adam13 Knockdown (KD), which reduces the adam13 protein to undetectable levels (*Figure 2C,A13* Pro-P and Mature-M) reduced the expression of pcdh8l mRNA by 43% (*Figure 2A*). To confirm this data, we injected MO13 (5 ng) into one blastomere of two-cell-stage embryos. The embryos were then fixed and stained by in situ hybridization with a pcdh8l probe. We observed that at stage 20, 71% of the embryos showed a reduced expression of pcdh8l on the injected side when compared to the non-injected control side (*Figure 2B*, N = 31). We also tested the protein level of pcdh8l by western blot, using mAb 2F4 (*Figure 2C*). When compared to non-injected embryos, the level of pcdh8l protein was significantly reduced. Because both adam13 and pcdh8l are also expressed in the somites, we tested if the reduction of pcdh8l was detectable in isolated CNC explants. CNC isolated from embryos injected with MO13 had a 36% reduction of pcdh8l mRNA compared to control CNC (*Figure 2—figure supplement 1A*). Again, western blot analysis with 2F4 showed a similar trend for the pcdh8l protein (*Figure 1—figure supplement 1B*). These results show that loss of adam13 results in a significant decrease of pcdh8l expression.

To test if adam13 could induce pcdh8l, we injected mRNA encoding either the wild-type adam13 or various mutant forms of the protein at the one-cell stage and dissected the naïve ectoderm (animal cap, AC) at stage 9. The animal cap explants were then grown until the control embryos reached neurula stage (17 to 20) and mRNA and protein expression were assessed by real-time PCR and western blot respectively. Analysis of mRNA levels show that wild-type adam13 induces pcdh8l

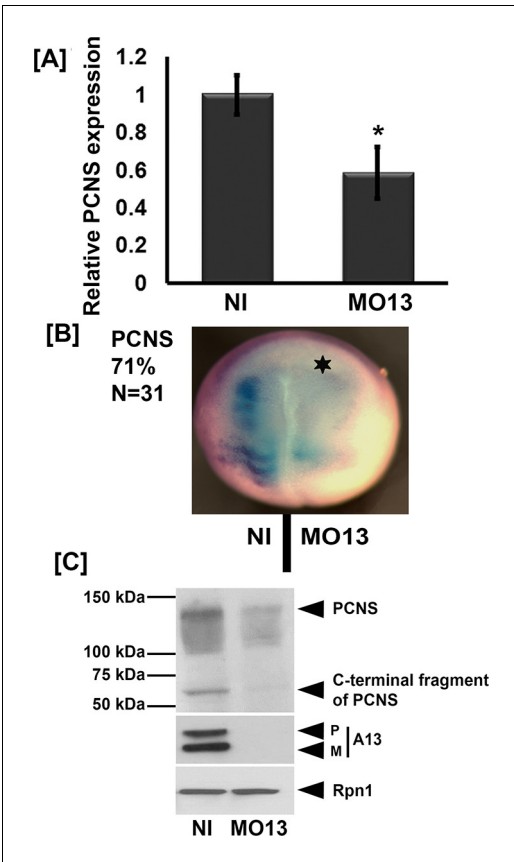

**Figure 2.** Adam13 knockdown reduces pcdh8l expression. (**A**) Relative mRNA expression of pcdh8l normalized to GAPDH obtained by real-time PCR. MO13 was injected at the one cell stage (10 ng) and embryos were collected at stage 20. Polyadenylated mRNA was extracted from non-injected and MO13 injected embryos (five embryos each). (**B**) Representative images of whole mount in situ hybridization in which MO13 (5 ng) was injected into one of the two blastomeres and non-injected side serves as control. Image shows reduced mRNA levels of pcdh8l in MO13 injected side (star, 71% of 31 embryos). (**C**) Western blot on glycoproteins isolated from either control (Non Injected) or knock down (10 ng MO13) stage 20 embryos (25 embryos each). MO13 efficiently prevents the translation of adam13 (**A13**) as well as reduces the pcdh8l protein level. A monoclonal antibody to Xenopus Ribophorin 1 (Rpn1) was used as a loading control. Error bars represent standard error to the mean (Mean ±S.E.M). One-way ANOVA was performed to determine statistical significance. Statistically significant at *$p<0.05$.

DOI: https://doi.org/10.7554/eLife.26898.006

The following source data and figure supplements are available for figure 2:

**Source data 1.** Source data for *Figure 2*.
DOI: https://doi.org/10.7554/eLife.26898.009

**Figure supplement 1.** adam13 knockdown reduces pcdh8l expression in cranial neural crest (CNC) cells.

*Figure 2 continued on next page*

expression whereas its catalytically inactive form (A13E/A), a mutant lacking the cytoplasmic domain (ΔCyto) and the isolated cytoplasmic domain (C13) all fail to induce expression of pcdh8l (*Figure 3A*). These results were confirmed at the level of pcdh8l protein expression using mAb 2F4 (*Figure 3B*). These results show that adam13 can induce the expression of pcdh8l in naïve ectoderm and that both the proteolytic activity and the cytoplasmic domain are required for this function.

## Adam13 regulates tfap2α expression

We have previously shown that adam13 can regulate multiple genes including the cytoplasmic protease Capn8a (*Cousin et al., 2011*). While we showed that the cytoplasmic domain of adam13 is cleaved and translocates into the nucleus, there are no known DNA-binding motifs or transactivation domains in its sequence. For this reason we tested whether adam13 could regulate the expression of a transcription factor known to activate the expression of Cpn8a and pcdh8l. Tfap2α is such a transcription factor, and it plays a key role in neural plate border specification where adam13 is expressed prior to neural crest migration (*de Crozé et al., 2011*; *Luo et al., 2005*; *Alfandari et al., 1997*). As described above for pcdh8l, we used the morpholino to adam13 to inhibit adam13 translation in embryos and collected them at stage 20. The real-time PCR data indicate that MO13 reduces tfap2α mRNA expression to 67% of control embryos (*Figure 4A*). In situ hybridization with a tfap2α probe shows that loss of adam13 reduces tfap2α mRNA expression on the injected side in 76% of the embryos (*Figure 4B*). To test if adam13 could regulate the tfap2α promoter, we cloned 2.5 Kbp of the *Xenopus laevis* genomic sequence upstream of the transcription start of the tfap2α gene into a luciferase vector (pGlo3, Bio-rad). We then injected at the 8-cell-stage in the dorsal animal blastomere the tfap2α:luciferase plasmid together with a Renilla control plasmid in both control embryos and embryos lacking adam13. Analysis of the luciferase activity in dissected CNC explants shows a significant reduction of the tfap2α promoter activity following adam13 KD (*Figure 4C*). This decrease is nearly identical to the observed reduction of the endogenous tfap2α mRNA (*Figure 4A*) suggesting that this reporter contains the key responsive elements regulated by adam13.

*Figure 2 continued*

DOI: https://doi.org/10.7554/eLife.26898.007

**Figure supplement 1—source data 1.** Source data for *Figure 2—figure supplement 1*.

DOI: https://doi.org/10.7554/eLife.26898.008

and cultured until sibling embryos reached neurula stage (17 to 20) to isolate mRNA for real-time PCR. The results show that compared to non-injected control (NI), animal caps injected with adam13 RNA have a significant increase in tfap2α mRNA expression levels (*Figure 5A*). Similar to what was observed for pcdh8l, both the cytoplasmic domain and the proteolytic activity of adam13 are required for the full expression of tfap2α.

We then used the tfap2α:luciferase construct to test if adam13 could induce tfap2αexpression in Human Hek293T cells. As seen in *Figure 5B*, transfection of adam13 results in an almost 4-fold increase in tfap2α promoter activity. We previously showed that the cytoplasmic domains of adam13 and 19 could rescue CNC migration in embryos expressing adam13 lacking a cytoplasmic domain (ΔCyto), while the cytoplasmic domain of adam9 could not. In addition both adam13 and 19 cytoplasmic domains could rescue the expression of Cpn8a in CNC while adam9 could not (*Cousin et al., 2011*). We therefore tested the ability of adam9, 13 and 19 to induce the tfap2α promoter (*Figure 5B*). As expected, only adam13 and adam19 could induce the expression of tfap2α while adam9 could not. In addition, the cytoplasmic domain of adam13 was critical for tfap2α induction. As seen for pcdh8l, the cytoplasmic domain alone was not able to induce tfap2α expression. In contrast to what was seen in the animal cap, the proteolytically inactive adam13 (A13E/A) mutant was able to induce the reporter even if less efficiently than the wild type adam13.

## Adam13 regulation of pcdh8l depends on tfap2α

To test if tfap2α was involved in adam13 induction of pcdh8l, we expressed adam13 in animal cap with or without knocking down tfap2α. As seen before, injection of adam13 results in an increase in expression of pcdh8l and tfap2α mRNA (*Figure 5C and D*). In contrast, injection of a tfap2α morpholino prevented the induction of both pcdh8l and tfap2αby adam13 (*Figure 5C and D*). In *Xenopus tropicalis*, adam13 has been shown to positively regulate the Wnt signaling pathway (*Wei et al., 2010*) and Wnt/β-catenin has been shown to regulate tfap2α. To test if adam13 control of pcdh8l or

## The adam13 cytoplasmic domain is critical for tfap2α induction

Given that adam13 is required for proper tfap2α expression, we then tested if adam13 expression in naïve ectoderm could induce tfap2αexpression. Again, we injected embryos at the one cell-stage with the various constructs. Animal caps were dissected at the blastula stage

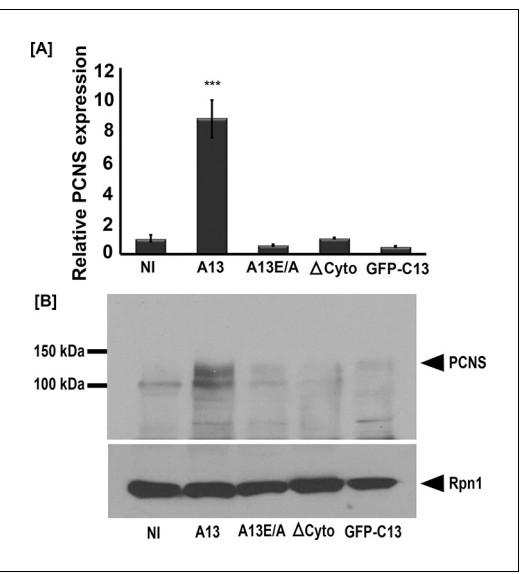

**Figure 3.** Full-length adam13 induces pcdh8l expression in naïve ectoderm (animal cap cells). One-cell stage embryos were injected with mRNA encoding various forms of adam13 (1 ng). Animal cap (AC) explants were dissected at stage 9 and grown in 0.5X MBS until sibling embryos reached stage 18 to 20. mRNA and protein was extracted from AC for analysis of pcdh8l. (**A**) Quantitative real-time PCR from mRNA isolated from 10 animal caps. The relative level of pcdh8l expression was normalized to GAPDH. (**B**) Western blot for pcdh8l protein using mAb2F4. 30 AC were used to extract protein from embryos injected with various adam13 constructs. Western blot shows induction of pcdh8l by full-length adam13, whereas other constructs fails to induce pcdh8l expression. adam13 (A13), non proteolytic adam13 (A13E/A), adam13 lacking a cytoplasmic domain (ΔCyto), isolated adam13 cytoplasmic domain (GFP-C13). Error bars represent standard error to the mean (Mean ±S.E.M). One-way ANOVA was performed to determine statistical significance. Statistically significant at ***p<0.001.

DOI: https://doi.org/10.7554/eLife.26898.010

The following source data is available for figure 3:

**Source data 1.** Source data for *Figure 3*.

DOI: https://doi.org/10.7554/eLife.26898.011

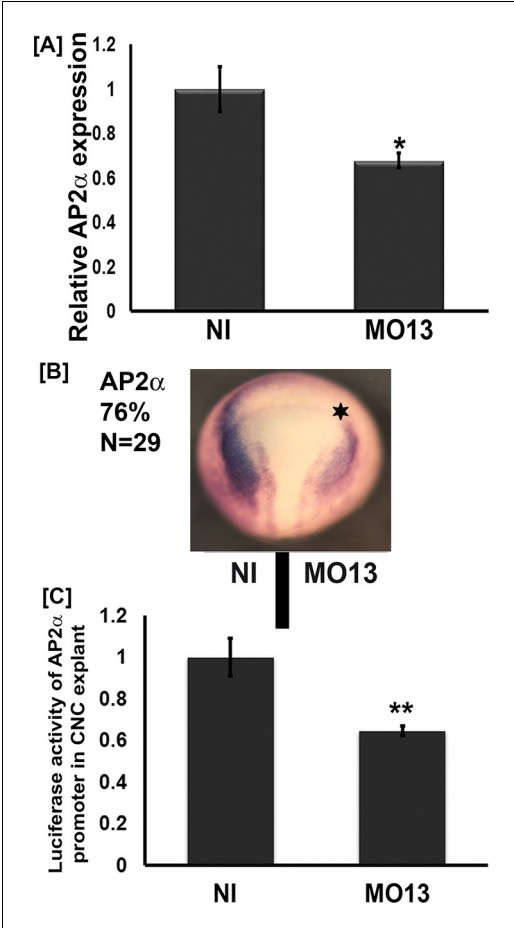

**Figure 4.** Adam13 knockdown reduces tfap2α expression. (**A**) Relative mRNA expression of AP2α normalized to GAPDH (five embryos). Real-time PCR analysis shows a 40% decrease in AP2α expression in response to adam13 KD (**B**) Representative images of dorsal view of whole mount in situ hybridization with AP2α. MO13 (5 ng) was injected in one of the two blastomeres (star). The non-injected side serves as control. adam13 KD reduces AP2α expression in 76% of the embryos (N = 29). (**C**) Luciferase activity of AP2α promoter in cranial neural crest cells (CNC). One-cell stage embryos were injected with MO13 (10 ng). Control and KD embryos were further injected at the 8 cell stage with the AP2α:luciferase reporter (100 pg) together with pRL-CMV (10 pg) in one animal dorsal blastomere. CNC explants were dissected from stage 15–17 embryos and individual explants were lysed to measure luciferase activity. The normalized values from 10 individual explants (two biological replicates) were used for each measure. Error bars represent standard error to the mean (Mean ±S.E.M). One-way ANOVA was performed to determine statistical significance. Statistically significant at *p<0.05, **p<0.01.

DOI: https://doi.org/10.7554/eLife.26898.012

The following source data is available for figure 4:

**Source data 1.** Source data for *Figure 4*.
DOI: https://doi.org/10.7554/eLife.26898.013

tfap2α also involves this pathway, we injected embryos with a morpholino to β−catenin (*Heasman et al., 2000*). The results show that reduction of β-catenin does not prevent adam13 induction of pcdh8l or tfap2α(*Figure 5C and D*). Interestingly adam13 up-regulation of tfap2α was also inhibited by the morpholino to tfap2α suggesting a positive feedback loop mechanism in which, tfap2α is involved in its own regulation by adam13 (*Figure 5D*).

## Adam13 physically interacts with arid3a and FoxD3

Our results suggest that adam13 can directly regulate the tfap2α promoter. However, there is no evidence that the cytoplasmic domain of adam13 can interact with DNA. In addition, fusing the adam13 cytoplasmic domain with the VP-16 trans-activator domain does not induce the expression of tfap2α (Alfandari, unpublished result). Taken together, these results suggest that adam13 may interact with and regulate a transcription factor that controls tfap2α expression. To test this hypothesis, we narrow down the list of potential candidate transcription factors based on the presence of their binding site on the tfap2α promoter (Jaspar [*Sandelin et al., 2004*]), their expression in the CNC (Xenbase, [*Bowes et al., 2010*; *Karpinka et al., 2015*]), and their ability to induce the tfap2α luciferase reporter in Hek293T cells (data not shown). We then tested whether any of these transcription factors could physically interact with adam13. Two transcription factors (foxd3 and arid3a) were positive in all assays and were tested for adam13 binding by co-immunoprecipitation (*Figure 6*). When injected into embryos, arid3a-flag co precipitated with adam13 (*Figure 6A*). This interaction was lost when adam13 was KD (MO13). Similarly when foxd3-myc RNA was injected in embryos either alone or with the adam13 MO, the foxd3 protein was co-precipitated with the endogenous adam13 protein and was absent when adam13 was KD (*Figure 6B*). These results show that both transcription factors can interact with adam13 in vivo.

## Arid3a is critical for tfap2α and pcdh8l expression

We then tested whether foxd3 and arid3a were required for adam13 induction of tfap2α and pcdh8l using the animal cap assay (*Figure 7A–B*). The real-time PCR shows that while arid3a KD efficiently prevented adam13 induction of both tfap2α and pcdh8l, foxd3 KD had no effect. In



**Figure 5.** Adam13 induces pcdh8l via AP2α. (A, C, D) Relative expression of AP2α and pcdh8l in naïve ectoderm (animal cap) by real-time PCR. One-cell stage embryos were injected with 1 ng of the various adam13 constructs and morpholinos and embryos were collected at stage 9 to dissect animal cap cells (AC). AC were grown in 0.5X MBS until sibling embryos reached stage 20 to 22. Messenger RNA was extracted from 10 AC for gene expression analysis. Expression was normalized using GAPDH and compared to the expression in non-injected AC (NI). (A) Real-time PCR data show that adam13 can induce AP2α by more than four fold. Both proteolytic activity and the presence of the cytoplasmic domain are essential for full activation. (B) Luciferase activity of AP2α promoter in Hek293T cells shows induction by adam13 and ADAM19 but not ADAM9. For each transfection the luciferase values were normalized to the Renilla values driven by the CMV promoter. In these assays, the absence of proteolytic activity (A13E/A) reduced AP2α induction only slightly, while the deletion of the cytoplasmic domain (ΔCyto) prevented the activity. (C) Induction of AP2α by adam13 was prevented by the KD of AP2α (10 ng of MOAP2α) but not β-catenin (20 ng of Moβ-catenin). (D) Expression of AP2α in response of adam13 also depends on AP2α but not β-catenin. Error bars represent standard error to the mean (Mean ±S.E.M). One-way ANOVA was performed to determine statistical significance. Statistically significant at **p<0.01, ***p<0.001.

DOI: https://doi.org/10.7554/eLife.26898.014

The following source data is available for figure 5:

**Source data 1.** Source data for *Figure 5*.
DOI: https://doi.org/10.7554/eLife.26898.015

these experiments, loss or gain of adam13 function had no effect on arid3a and foxd3 expression (*Figure 7—figure supplement 1*).

Because arid3a was shown to regulate TGFβ signaling downstream of Smad2 (*Callery et al., 2005*) and the ability of mammalian ADAM12 to mediate TGFβ signaling (*Ruff et al., 2015*), we investigated whether adam13 induction of pcdh8l and tfap2α requires Smad2. Animal cap experiments show that adam13 induces tfap2α and pcdh8l expression even when Smad2 is knocked down (*Figure 8A–B*). In addition Smad2 does not induce the tfap2αpromoter in Hek293T cells and the

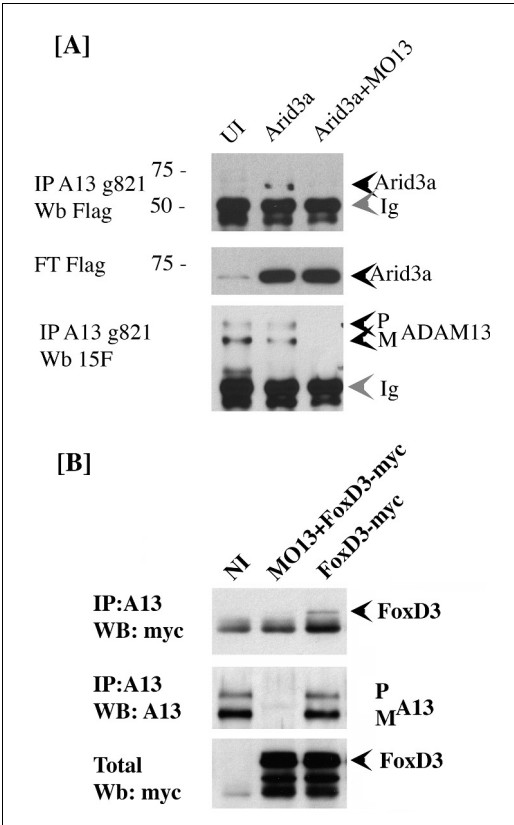

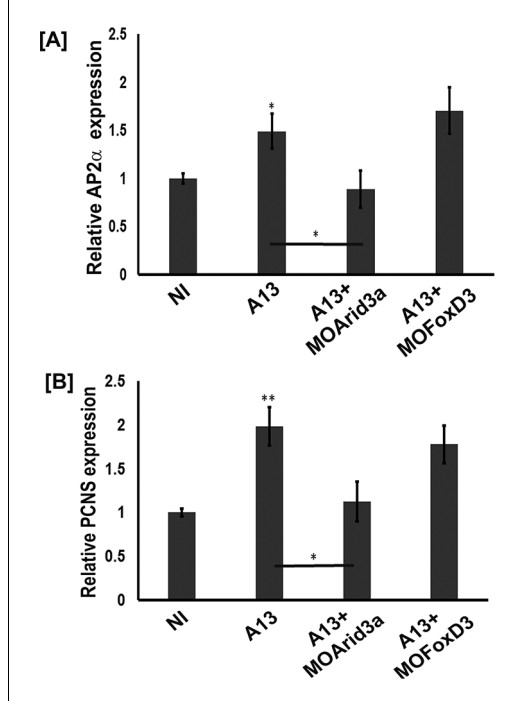

**Figure 6.** adam13 binds to arid3a and foxd3. (**A**) Co-immunoprecipitation of arid3a-flag with adam13. Arid3a-flag mRNA were injected in one-cell stage embryos either alone or with the morpholino to adam13 (MO13.) Proteins were immunoprecipitated with a goat polyclonal antibody directed against the cytoplasmic domain of adam13 (g821, [*Cousin et al., 2011*]) and blotted with M2 to detect the Flag-tag of arid3a. The flow through was used to detect arid3a. The immunoprecipitation were re-probed using 6615F to detect adam13 (*Alfandari et al., 1997*). (**B**) Co-immunoprecipitation of adam13 and FoxD3 from embryos. FoxD3-myc mRNA was injected at the one-cell stage either alone or with MO13. Adam13 was immunoprecipitated using the goat polyclonal antibody to adam13 (g821), and the proteins were detected by western blot using either the myc antibody (9E10) or a rabbit antibody to adam13 (6615F). FoxD3 co-precipitated with endogenous adam13.
DOI: https://doi.org/10.7554/eLife.26898.016

**Figure 7.** adam13 requires arid3a for induction of AP2α. Relative expression of AP2α (**A**) and pcdh8l (**B**) in animal caps from embryos injected with adam13 alone or with a morpholino to arid3a (20 ng) or FoxD3 (20 ng). Animal caps were extracted at stage 17.
DOI: https://doi.org/10.7554/eLife.26898.017

The following source data and figure supplements are available for figure 7:

**Source data 1.** Source data for *Figure 7*.
DOI: https://doi.org/10.7554/eLife.26898.020
**Figure supplement 1.** adam13 does not induce arid3a and FoxD3 expression in Naïve ectoderm.
DOI: https://doi.org/10.7554/eLife.26898.018
**Figure supplement 1—source data 1.** Source data for *Figure 7—figure supplement 1*.
DOI: https://doi.org/10.7554/eLife.26898.019

inhibitory Smad, Smad7 does not prevent adam13 induction of the tfap2α promoter in the same cells (*Figure 8C*). These results suggest that adam13 regulation of tfap2α does not involve the TGFβ signaling pathway.

While arid3a has been shown to be essential for mesoderm specification (*Callery et al., 2005*), there is no evidence concerning its role in neural crest cell induction or patterning. Published expression pattern shows that it is expressed in the non-neural ectoderm and the neural plate border (*Callery et al., 2005*) where it overlaps with adam13 suggesting that it could play a role in neural crest specification. We, therefore, investigated the effect of the loss of arid3a on the expression of tfap2α, pcdh8l and the neural crest marker snai2. To avoid a general defect in mesoderm patterning, we injected embryos at the 8-cell stage in a single dorsal animal blastomere and tested the expression of the selected genes by in situ hybridization (*Figure 9A–C*). The results show that loss of arid3a

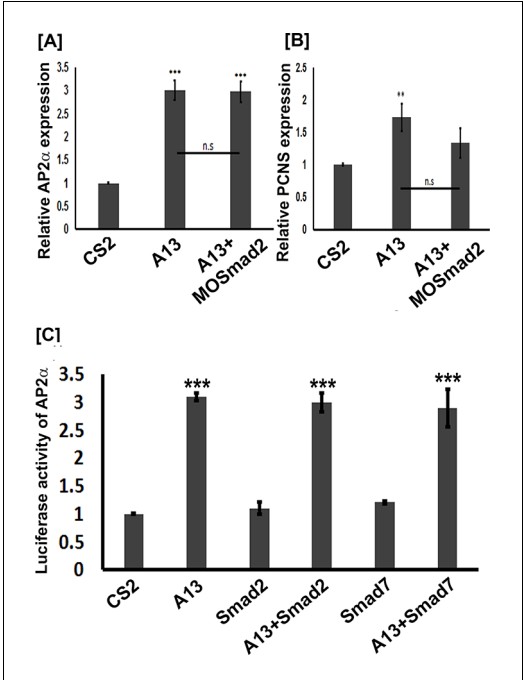

**Figure 8.** adam13 induction of Ap2α and pcdh8l does not involve Smad2. (**A–B**) Relative expression of AP2α and pcdh8l in animal caps dissected from control embryos, or embryos injected with adam13 or adam13 and a morpholino to Smad2 (MOSmad2, 25 ng). (**C**) Luciferase assays in Hek293T cells show no induction of AP2α promoter activity by Smad2 (0.5 μg) or Smad 7 (0.5 μg). Smad2 does not increase adam13 activation of the AP2α promoter, while Smad7 does not reduce adam13 induction of AP2α promoter. Smad2 and Smad7 were transfected with or without adam13 (0.5 μg) along with AP2α promoter-luciferase and the pRL-CMV (100 ng, 10 pg). The ratio of luciferase to renilla was used to normalize each transfection to the empty vector control (CS2). Error bars represent standard error to the mean (Mean ± S.E.M). One-way ANOVA was performed to determine statistical significance. Statistically significant at **p<0.01, ***p<0.001.
DOI: https://doi.org/10.7554/eLife.26898.021
The following source data is available for figure 8:

**Source data 1.** Source data for *Figure 8*.
DOI: https://doi.org/10.7554/eLife.26898.022

decreased the expression of tfap2α and pcdh8l as expected but also snai2, suggesting a key role in neural crest cell patterning.

## Tfap2α but not arid3a can rescue CNC migration in embryos lacking adam13

It is clear that adam13 regulates tfap2α expression, and therefore, it is expected that restoring the level of tfap2α in embryos lacking adam13 should at least partially rescue CNC migration. Indeed, injection of tfap2α in embryos lacking adam13 significantly rescued CNC migration (*Figure 9E–F*) in a way similar to that of pcdh8l expression. In contrast, expression of arid3a had no effect on CNC migration in embryos lacking adam13. These results are consistent with a model where adam13 requires arid3a to function in the CNC. The facts that arid3a expression is not regulated by adam13 (*Figure 7—figure supplement 1*), that both proteins interact (*Figure 6*) and that arid3a cannot compensate for the loss of adam13 (*Figure 9E*) suggest that arid3a is not downstream of adam13 but rather that the two proteins work together.

## Adam13 stimulates Arid3a nuclear translocation

To test if adam13 could regulate Arid3a translocation to the nucleus, we performed nuclear and cytoplasmic fractionation of transfected Hek293T cells (*Figure 10*). As expected we found that arid3a is equally distributed between the cytoplasmic and nuclear compartment (*Figure 10A*). Interestingly, in the nucleus a smaller fragment (40 kDa) is observed, and this fragment is much more abundant in the nucleus of cells co-transfected with adam13. We then tested if this fragment was also observed in cells co-transfected with the adam13 protease dead mutant (A13E/A) and the mutant lacking the cytoplasmic domain (A13ΔCyto). In both conditions, we found that the fragment was not increased by these mutants (*Figure 10B*). We then tested if Arid3a was localized to the plasma membrane in Hek293T cells, as previously described for B-cells, and if adam13 and the various mutants affected this localization (*Figure 10C*). Transfected Arid3a was found in the membrane fraction of Hek293T cells in all conditions tested. In cells expressing adam13, the 40 kDa fragment of arid3a (arid3a Short) was enriched. This fragment was dramatically increased in cells expressing the A13E/A mutant, suggesting that the proteolytic activity of adam13 is not responsible for the production of arid3a Short. In contrast, cells expressing A13ΔCyto express the same level of arid3a Short than those expressing only arid3a. Taken together, these results show that adam13 regulates arid3a post-translational modification (proteolytic cleavage) and nuclear translocation. These results are compatible with a model (*Figure 11-1*) in which the adam13 cytoplasmic domain is part of a complex that either cleaves arid3a at

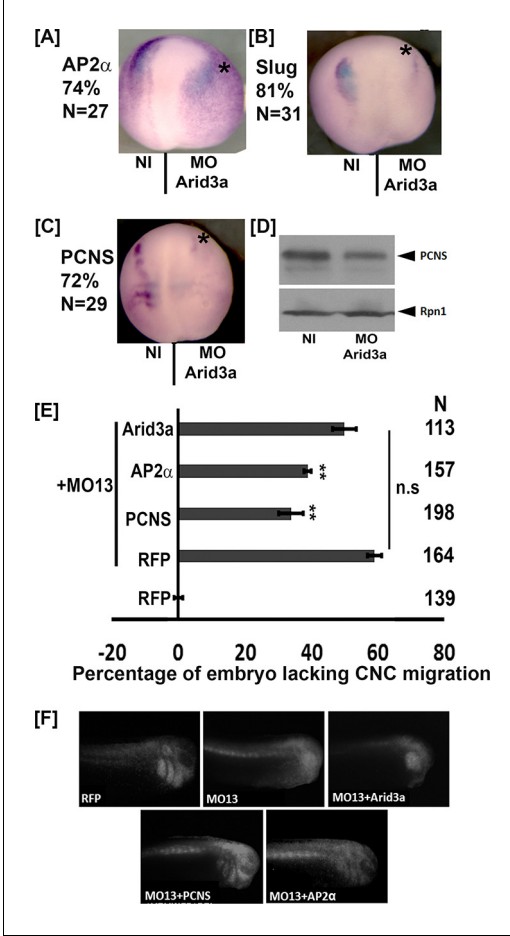

**Figure 9.** arid3a is critical for multiple gene expression in the CNC. (**A–C**) Representative dorsal view of neurula treated by whole mount in situ hybridization with probes for tfap2α (AP2α), snai2 (Slug) and pcdh8l (PCNS). Eight-cell stage embryos were injected in one dorsal animal blastomere with the arid3a morpholino (5 ng, Asterisk). The percentage of embryos with reduced signal in the injected side is given in N is the total number of embryos obtained from each case. (**D**) Western blot using mAb2F4 to detect pcdh8l (PCNS). Glycoproteins from 25 embryos were extracted and purify on ConA-agarose beads. Rpn1 was used as a loading control. (**E**) Histogram representing the percentage of embryos lacking CNC migration. One-cell stage embryos were injected with MO13. Control or KD embryos were further injected at the 8 cell stage in one animal dorsal blastomere with mRNA for RFP, PCNS, AP2α or Arid3a. Observation of the RFP fluorescence at stage 24–26 reveals that the inhibition of migration by adam13 KD is partially rescued by pcdh8l and AP2α but not Arid3a. Error bars represent standard error to the mean (Mean ± S.E.M). One-way ANOVA was performed to determine statistical significance. Statistically significant at *p<0.05, **p<0.01. (**F**) Representative examples of injected embryos. Posterior to the left, dorsal is up.
DOI: https://doi.org/10.7554/eLife.26898.023

*Figure 9 continued on next page*

the plasma membrane or stabilizes a cleaved form of arid3a (*Figure 11-2*). Upon adam13 self-proteolysis (*Cousin et al., 2011*; *Gaultier et al., 2002*) (*Figure 11*), gamma secretase can cleave the cytoplasmic domain of adam13 (*Figure 11-3*), which then translocates with or without arid3a to the nucleus (*Figure 11-4*). In the absence of adam13 proteolytic activity, the complex remains stable at the plasma membrane (*Figure 10C*), while in the absence of the cytoplasmic domain the complex is not formed.

## Discussion

Our results show that adam13 expression is essential for the proper expression of tfap2α, a transcription factor that defines the neural plate border in all vertebrates studied so far (*de Crozé et al., 2011*; *Hoffman et al., 2007*; *Luo et al., 2003*; *Martinelli et al., 2011*; *Van Otterloo et al., 2012*; *Van Otterloo et al., 2010*). This regulation is mediated by the interaction of adam13 with the arid3a transcription factor and requires both the cytoplasmic domain and metalloprotease activity of adam13 in naïve ectoderm. Our results suggest that the cytoplasmic domain of adam13 is critical for either stabilizing or inducing the cleavage of arid3a at the plasma membrane. This fragment is then accumulated in the nucleus of cells co-transfected with adam13. The accumulation of this fragment in the nucleus appears to be controlled by the adam13 proteolytic activity, as it is impaired in the A13E/A mutant. Given adam13's ability to cleave itself within its cysteine rich domain (*Gaultier et al., 2002*), thus becoming a substrate for gamma-secretase, to cleave and release the cytoplasmic domain from the membrane (*Cousin et al., 2011*), we propose that the complex containing the cleaved arid3a, and at least the adam13 cytoplasmic domain is then free to translocate into the nucleus (*Figure 11*). This model is consistent with the essential roles of both the proteolytic and cytoplasmic domain of adam13 in animal cap and the CNC. It is also consistent with the observation that the GFP-C13 fusion protein containing the adam13 cytoplasmic domain, which is found only in the nucleus, is unable to induce tfap2α expression. On the other hand, the fact that the adam13 E/A mutant can induce the reporter in Hek293T cells even at a lower efficiency is puzzling. While this construct clearly increases the arid3a fragment, it does not allow its efficient nuclear translocation (*Figure 10B and C*). One possible explanation is that it may act as a dominant mutant by trapping e2f1 at the plasma

*Figure 9 continued*

The following source data is available for figure 9:

**Source data 1.** Source data for *Figure 9*.
DOI: https://doi.org/10.7554/eLife.26898.024

membrane. Arid3a was originally identified in a screen of E2F1 binding protein as E2FBP1 (*Suzuki et al., 1998*). In Hek293T cells, e2f1 inhibits the tfap2a promoter activity and is able to block both adam13 and arid3a induction of the reporter (data not shown). Thus by binding to the endogenous E2F1 protein in Hek293T cells, A13E/A could sequester E2F1 at the plasma membrane and increase the reporter activity. In Xenopus, e2f1 is expressed maternally and is strongly down regulated at stage 20 (*Owens et al., 2016*) when the animal caps are extracted. Thus, A13E/A expression would have no effect on its ability to translocate in the nucleus.

In B-cells, ARID3a also known as Bright, is post translationally modified. First, it is palmitoylated and associate with lipid raft at the plasma membrane (*Schmidt et al., 2009*). Upon stimulation of the B-cell receptor, arid3a is sumoylated, dissociate from the lipid raft and is then free to regulate gene expression (*Prieur et al., 2009*). It is possible that in the neural crest cells, adam13 regulates a pool of arid3a associated with lipid raft at the plasma membrane. It could do this by helping assemble the proper signaling complex including the enzyme responsible for arid3a post-translational modification. In the complex, the presence of a protease would cleave off the N-terminal domain of arid3a. This cleavage could be important for releasing a transcriptional inhibitor (e.g. e2f1) or simply destabilizing arid3a interaction with the plasma membrane. Given the widespread expression of arid3a, this would provide a mechanism by which the transcription factor is only activated in the right cells and at the right time. Our results show that adam13 expression dramatically increases the nuclear localization of a cleavage product of arid3a (arid3a Short), while it does not affect the overall distribution of the intact arid3a protein. This depends on both the proteolytic activity and the cytoplasmic domain of adam13. It is tempting to speculate that this fragment (arid3a Short), which has not been described before, is responsible for the increased expression of tfap2α. Based on the calculated molecular weight of the fragment, it is likely to maintain the intact ARID (DNA binding) domain as well as the RECKLES domain that control the nuclear localization. It will be interesting to determine if adam13 stabilizes arid3a Short or produces it. If this model were correct, we would expect that proteins that interfere with the production of the cleaved cytoplasmic domain would also interfere with adam13's ability to induce tfap2α. We have previously shown that the secreted Wnt receptor Fz4-v1, binds to adam13 and prevents its proteolytic activity. We further showed that KD of Fz4-v1 induces an early production of the adam13 cytoplasmic fragment (*Abbruzzese et al., 2015*). In embryos, KD of Fz4-v1, increases both tfap2α and pcdh8l transcription (data not shown), while In Hek293T cells, expression of Fzd4-v1 inhibits adam13 activation of the tfap2α reporter further supporting our model.

Our results also show that adam13 activation of tfap2α in naïve ectoderm does not require β-catenin, while tfap2α has been shown to be induced by β-catenin in the CNC (*de Crozé et al., 2011*). In addition, we also show that adam13 induction of tfap2α requires the presence of tfap2α. Both adam13 and tfap2α are initially expressed in the entire ectoderm and become excluded from the neural plate to be restricted to the neural folds and later neural crest (*Alfandari et al., 1997*; *Luo et al., 2002*). In addition, the quantitative expression pattern of tfap2α and adam13 by RNA-seq is nearly identical (*Owens et al., 2016*). This suggests that adam13 and tfap2αexpression is initially controlled by the same transcription factor possibly β-catenin (*Deardorff et al., 2001*), and that adam13 is later required for the maintenance and amplification of tfap2αexpression within the neural fold and neural crest cells. This is compatible with the fact that adam13 is not required for β-catenin induction of snai2 in the CNC in *Xenopus laevis*. Also consistent with this model, we find that both arid3a and tfap2α can induce the tfap2α reporter individually, and the combination of these two proteins produces an additive increase in the luciferase expression (data not shown). We also tested if adam13 could initiate a full CNC program in naïve ectoderm. We found that while snail2 and twist can be induced by adam13, cadherin 11, sox8 and sox10 were not (data not shown) suggesting that adam13 is not capable to induce neural crest cells from naïve ectoderm.

Ultimately, adam13 induction of tfap2α results in the induction of pcdh8l and the up-regulation of Cpn8a in the CNC, two proteins critical for CNC migration. The fact that CNC lacking adam13 can migrate in vitro while those missing pcdh8l cannot (*Rangarajan et al., 2006*) might be explained by the observation that the CNC lacking adam13 eventually expresses both tfap2α and pcdh8l

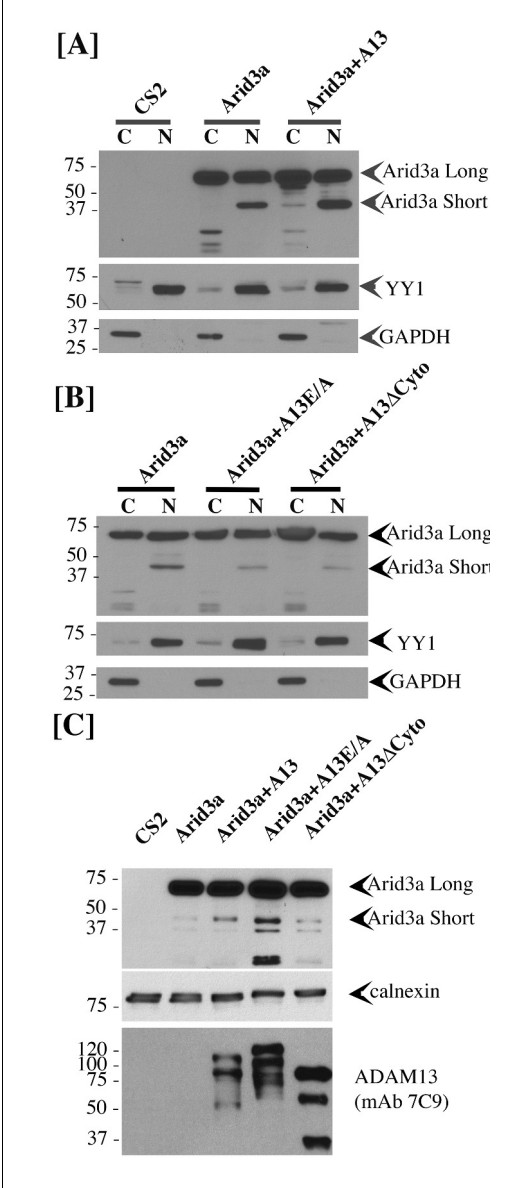

**Figure 10.** adam13 regulates arid3a post-translational modification. Western blot from transfected Hek293T cells. (**A–B**) cytoplasmic (**C**) and nuclear (**N**) extracts from cells transfected with the empty vector (CS2), Arid3a-flag, adam13 (A13) or both. The blots were re-probed with the transcription factor YY1 as a nuclear marker and GAPDH as a cytoplasmic marker. The full-length *Xenopus laevis* Arid3a is detected at approximately 60 kDa (Arid3a Long). A shorter fragment is detected at about 40 kDa (Arid3a Short). (**A**) Co-transfection of adam13 with Arid3a increases the Arid3a protein level in the cytoplasm by 30% and the shorter fragment of Arid3a in the nucleus by five folds. (**B**) Co-transfection of Arid3a with the proteolytically inactive mutant adam13 (A13E/A) or the mutant lacking the cytoplasmic domain (A13ΔCyto) does not increase the shorter fragment in the nucleus. (**C**) Membrane extract from Hek293T cells transfected with Arid3a-flag

suggesting that another control of tfap2α expression is activated later on. This may or may not require the involvement of other ADAM proteases. In particular, it is possible that adam19, which can activate the tfap2α reporter in Hek293T cells may compensate for the loss of adam13 in the CNC. It will be of interest to test if tfap2α expression in mouse is also regulated by ADAM proteases and if this contributes to craniofacial development. Given the conserved ability of the adam19 cytoplasmic domain from mouse and Xenopus to functionally replace the adam13 cytoplasmic domain, together with the ability of adam19 to induce the tfap2α reporter, it is tempting to speculate that adam19, which is expressed in mouse CNC and contributes to the cardiac neural crest cell development, could regulate tfap2α to promote CNC migration. While a role for adam13/33 and adam19 in mouse CNC has not been shown, it is possible that the craniofacial phenotypes are subtle and may have been missed in the individual Knock out reports.

Our results show that the extracellular fragment of multiple cadherins and protocadherin can rescue CNC migration. In particular, we have shown that the extracellular fragment of Cadherin-11 can rescue CNC migration in embryos lacking adam13 or expressing a non-cleavable form of cadherin-11 (**Abbruzzese et al., 2016**). Similarly, the extracellular fragment of pcdh8l partially rescues CNC migration in embryos lacking adam13. It is interesting to note that one can greatly improve the rescue either by injecting full-length pcdh8l together with adam13 lacking a cytoplasmic domain (data not shown). This mutant form of adam13 can cleave pcdh8l but cannot regulate transcription and therefore the level of the protocadherin. Alternatively, greater rescue is also achieved by expressing the EC1-4 fragment of pcdh8l together with tfap2α showing that the transcriptional and proteolytic activity of adam13 can be uncoupled. Why can the various cadherin and protocadherin fragments rescue migration? We have shown that the extracellular fragment of cadherin-11 bind to ErbB2 (EGF receptor 2) and regulates Akt phosphorylation (Mathavan et al., submitted). In cancer cells, the extracellular fragment of E-cadherin can also bind to ErbB2 to promote cell migration (**Brouxhon et al., 2014a**; **Brouxhon et al., 2014b**). This suggests that a common structure within the Cadherin/protocadherin extracellular domain can bind and signal through growth factor receptors.

*Figure 10 continued*

and the adam13 constructs. Co-transfection of Arid3a with adam13 increases the intensity of the 40 kDa Arid3a fragment. This is not observed in the absence of the adam13 cytoplasmic domain. In contrast, a much more significant increase is observed when Arid3a is co-transfected with the A13E/A mutant.

DOI: https://doi.org/10.7554/eLife.26898.025

## The larger role of ADAM regulation of transcription

Our results show how ADAMs can regulate transcription via their cytoplasmic domain. Multiple reports have shown ADAM protein in the nuclei (*Cousin et al., 2011*; *Friedrich et al., 2011*; *Arima et al., 2007*). Our previous study has shown that most ADAM cytoplasmic domains can translocate into the nucleus if cleaved from the plasma membrane. For adam13, we have identified a physical interaction with importin β1, which is known to mediate transport of proteins that do not harbor a canonical nuclear localization signal (Alfandari ms/ms unpublished). Given the essential and conserved role of tfap2α in craniofacial development (*de Crozé et al., 2011*; *Hoffman et al., 2007*; *Luo et al., 2003*; *Martinelli et al., 2011*; *Meshcheryakova et al., 2015*; *Van Otterloo et al., 2012*; *Hong et al., 2014*), as well as its ability to reduce the proliferation and migration of multiple cancer cells including those derived from neural crest cells (*Su et al., 2014*; *Li et al., 2010*; *Bennett et al., 2009*; *Orso et al., 2007*), it will be essential to study the potential regulation of tfap2α by human ADAM proteins. Our preliminary results show that mouse adam33 and adam12 are also able to induce the tfap2α reporter when transfected in Hek293T cells even if less efficiently than adam13 or adam19 while Xenopus adam9, 10 and 11 do not (data not shown). A better understanding of the precise regulation of tfap2α transcription by the various ADAMs will be necessary to determine if they all regulate gene expression by a common mechanism. Similarly, the observed role of adam12 in mediating TGF-β signaling, may

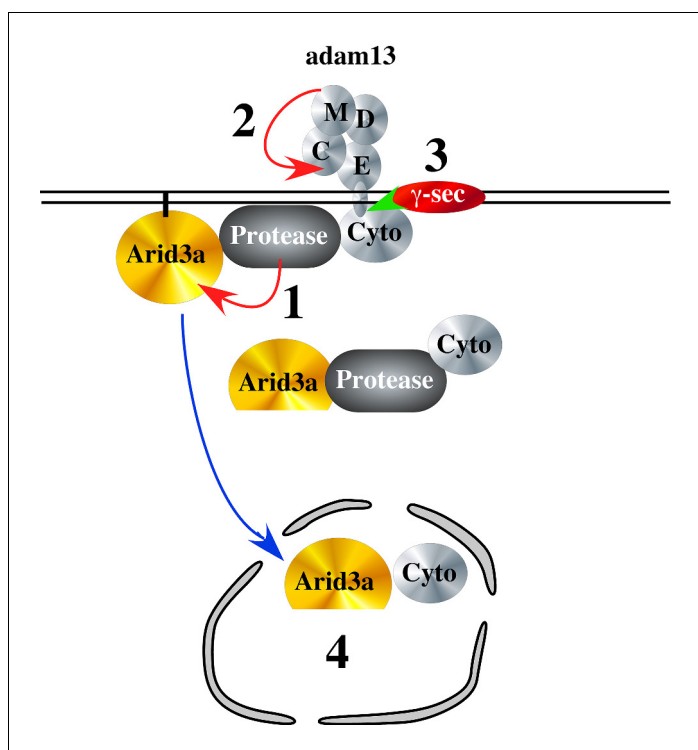

**Figure 11.** Hypothetical model of adam13 function: Adam13 at the membrane may associate with a cytoplasmic protease that cleaves Arid3a (1) to generate the short form seen in *Figure 10*. Arid3a is localized to lipid raft via its palmytoylation. Once adam13 cleaves itself in the cysteine rich domain (2), gamma secretase cleaves the cytoplasmic domain of adam13 (3), releasing the complex that can translocate in the nucleus to activate tfap2α transcription.

DOI: https://doi.org/10.7554/eLife.26898.026

involve an interaction with arid3a, a protein known to be downstream of the TGF-β effector Smad2 (*Callery et al., 2005*). The fact that both adam12 and arid3a are critical for trophoblast formation (*Rhee et al., 2017*; *Aghababaei et al., 2015*; *Rhee et al., 2014*; *Aghababaei et al., 2014*) suggests that this functional interaction may be important in multiple cell types to mediate key developmental processes.

# Materials and methods

## Antibodies

The monoclonal antibody to pcdh8l (mAb2F4, RRID:AB_2687672) was generated against a bacterial fusion protein corresponding to the entire cytoplasmic domain of pcdh8l using standard hybridoma techniques. It recognizes the endogenous protein by western blot and immunofluorescence. 2F4 does not recognize overexpressed protocadherin pcdh8 and pcdh1. The monoclonal antibody to Rpn1 (Mono5, RRID:AB_2687673) was generated by immunizing BalbC mice with Xenopus XTC cells (*Pudney et al., 1973*). Hybridomas were screened by ELISA on fixed Xenopus XTC cells (RRID: CVCL_5610) and western blot from embryo extracts. Mono5 was further characterized by shot-gun LC/ms/ms of protein immunoprecipitated using Mono5 from embryo extracts. The identity was confirmed by detecting the recombinant Xenopus Rpn1 transfected in Hek293T cells (ATCC, CRL-3216 RCB2202). All cell lines were treated with mycoplasma removal agent (MRA, MP-Biomedicals) prior to use. The antibody works by western blot and immunofluorescence on the endogenous Xenopus protein. The antibodies to adam13 have been described before. 6615F RRID:AB_2687671, rabbit polyclonal antibody (*Alfandari et al., 1997*), DC13, RRID:AB_2687674 Rabbit polyclonal antibody (*Gaultier et al., 2002*), gA13, RRID:AB_2687675 goat polyclonal antibody to the adam13 cytoplasmic domain (*Cousin et al., 2011*) mAb 4A7, RRID:AB_2687676 mouse monoclonal antibody to the cytoplasmic domain of adam13 (*Abbruzzese et al., 2014*). M2, mouse monoclonal antibody to Flag (Sigma, RRID:AB_439685). Rabbit polyclonal antibody to tfap2α (LS-C624, LifeSpan BioSciences, RRID:AB_2199419).

## Morpholinos and DNA constructs

Morpholino antisense oligonucleotides (Gene Tools, Philomath OR) were described elsewhere. MO13 (*McCusker et al., 2009*), MOtfap2α (*Luo et al., 2003*), MO FoxD3 (*Sato et al., 2005*), MO arid3a (*Callery et al., 2005*), MOpcdh8l (*Rangarajan et al., 2006*), 2006), MOSmad2 (*Rankin et al., 2011*), MOβ-catenin (*Heasman et al., 2000*). Full length pcdh8l and tfap2α in pCS2 were a generous gift from Dr. Thomas Sargent (NIH, Bethesda, MD, USA). The N-terminal flag tag was added in pcdh8l using the In-Fusion kit (Clontech, CA) according to manufacturer's instruction. The EC1-4 construct was produced by inserting a stop codon in the N-terminus tagged pcdh8l after the fourth cadherin repeat. All ADAM constructs are in pCS2 and have been described before. adam9 (*McCusker et al., 2009*; *Cai et al., 1998*). adam13, E/A-adam13, A13ΔCyto, GFP-C13 (*Cousin et al., 2011*; *Alfandari et al., 2001*), adam19 (*Neuner et al., 2009*). Monomeric red fluorescent protein (mRFP) in pCS2 was a generous gift from Dr. Jim Smith (Gurdon Institute, Cambridge, United Kingdom). All constructs were sequenced and transfected in Hek293T cells to verify the protein translation, size and surface expression when appropriate.

## Injections

Capped mRNAs were synthesized using SP6 RNA polymerase on DNA linearized with NotI as described before (*Cousin et al., 2000*). Injectors were calibrated using a 1 μl capillary needle (Microcaps, Drumond, PA, USA). The injection pressure was set at 15 psi and the injection time set between 50 and 200 ms to obtain a 5 nl delivery. For the CNC migration assays, the morpholino was injected at the 1 cell stage and mRNAs at 8 cell stage as described previously (*Abbruzzese et al., 2014*). Embryos were raised at 15°C until tail bud stage (St. 24 to 28) at which time CNC migration was scored for the presence or absence of fluorescent CNC cells in the migration pathways. For each injection the percentage of inhibition was normalized to embryos injected with RFP alone and set to 0% inhibition. All experiments were performed at least three times using different females to determine statistical significance. Each experiment always included a positive control (RFP) and MO13 in addition to all experimental conditions.

## Cell culture and transfection

Hek293T (ATCC, CRL-3216 RCB2202) cells were cultured in RPMI media supplemented with Pen/Strep, L-glutamine, sodium pyruvate, and FBS (10 U/ml, 2 mM; 0.11 mg/ml, 10%; Hyclone, South Logan, UT). Transfections were performed using X-tremeGENE HP DNA Transfection Reagent (Roche, Basel, Switzerland) following the manufacturer's instructions. For pcdh8l shedding assay, cells were incubated with media containing 2% serum for 24 hr after transfection and conditioned media was collected. The shed extracellular domain was purified using Concanavalin-A-agarose (Vector), eluted in reducing Laemmi and blotted using the Flag (M2) monoclonal antibody.

## Animal cap dissection

Embryos were injected at one cell stage with various mRNA constructs or morpholino (quantity provided in figure legends). At stage nine animal caps were dissected and grown in 0.5X Modified Barth's Saline (1XMBS: 88.0 mM NaCl, 1.0 mM KCl, 2.4 mM NaHCO$_3$, 15.0 mM HEPES [pH 7.6], 0.3 mM CaNO$_3$-4H$_2$O, 0.41 mM CaCl$_2$-6H$_2$O, 0.82 mM MgSO$_4$) with gentamicin (50 µg/ml) on agarose plate at 15°C until control embryos reached stage 17 to 20.

## Quantitative PCR

Quantitative real-time PCR was performed as previously described (*Neuner et al., 2009*). All primers were tested for efficiency. Embryos and explants were collected at the appropriate stage and immediately placed in a guanidinium thioisocyanate solution to extract RNA (Roche, RNA isolation Kit). Total RNA was quantified by absorbance at 260 nm using a nanodrop (Thermo Scientific). Polyadenylated RNA was isolated with oligo-dT beads (Qiagen) following manufacturer's instruction. The cDNA was produced using polyA mRNA purified from 10 CNC explants, or 10 animal caps or five embryos using direct cDNA synthesis kit (Quanta) according to manufacturer's instruction. Quantitative PCR was performed using SYBR green (Takara, Kyoto, Japan) to measure mRNA levels of pcdh8l, tfap2α and arid3a. GAPDH was used to normalize total cDNA quantities. Primer sequences are given in *Table 1*. The ΔΔCT (*Livak and Schmittgen, 2001*) technique was used to calculate fold changes. All results are presented as the fold change compare to non-injected.

## Immunoprecipitation and western blots

Hek293T cells, embryos and explants were all extracted in 1XMBS with 1% Triton-X100, protease phosphatase inhibitor cocktail (Thermoscientific) and 5 mM EDTA. Immunoprecipitations were performed using 1–5 µg of antibody bound to proteinA/G-agarose beads (Thermoscientific) either 2 hr at room temperature or overnight at 4°C. Beads were washed three times with the extraction buffer prior to elution with Laemmli buffer. All proteins were separated on 5–22% gradient SDS-PAGE gels and transferred to polyvinylidene fluoride membranes (PVDF, Millipore, Billerica, MA) using a semi-dry transfer apparatus (Hoeffer). pcdh8l glycoprotein was pulled down from total protein extract with concanavalin-A agarose beads (Vector Laboratories, Burlingame, CA). Cytoplasmic and nuclear extracts were performed using the NEper kit (Pierce) following manufacturer's instructions. Purification of total membrane was adapted from (*Tamura et al., 2011*), Hek293T cells were washed once

**Table 1.**

| Target | Sequence 5'−3' |
| --- | --- |
| xltfap2α Forward | ATAACAATGCGGTGTCGTCCCTCT |
| xltfap2α Reverse | AGAGCCTTCTCTGGACTTCTGCAA |
| xlarid3a Forward | GGAGGCTTGGTGGAAGTTATTA |
| xlarid3a Reverse | ACTGAGTGCGCAGAGTAAAG |
| xlGAPDH Forward | TTAAGACTGCATCAGAGGGCCCAA |
| xlGAPDH Reverse | GGGCAATTCCAGCATCAGCATCAA |
| xlpcdh8l Forward | TCTCAACTCGTGCTCAAATC |
| xlpcdh8l Reverse | CCTCTGCTGACCCATTATTC |

DOI: https://doi.org/10.7554/eLife.26898.027

with PBS and collected by pipetting up and down in ice cold EB (10 mM Hepes pH7.5, 0.3M sucrose, 1 mM EDTA). Cells were collected by spinning at 300 g for 5 min and were then extracted using multiple passages through a 25 g needle (1 ml for a single well of a six well plate) of EB buffer complemented with protease phosphatase inhibitor cocktail (Thermoscientific) and 5 mM EDTA. Intact cells and nuclei were pelleted at 1000 g for 10 min at 4°C. The supernatant were spun at 45,000 rpm/88,000 g (Beckman optima, TLA 100.2) for 30 min at 4°C. The membrane pellet was re-suspended in 100 μl of laemmly buffer. For embryos, 20 embryos were extracted in 500 μl of EB buffer, cell debris and nuclei were spun down at 1,000 g at 4°C for 15 min. The supernatant was then spun at 45,000 rpm/88,000 g at 4C for 30 min to pellet the total membranes.

## Whole mount in situ hybridization

Whole mount in situ hybridization was performed using previously described protocol (*Harland, 1991*). pcdh8l, tfap2α and snai2 probes were generated using diogoxigenin-rUTP–label. Images were taken using a Zeiss stereo microscope Lumar-V12 with the Axiovision software package.

## Luciferase activity for tfap2α promoter

Hek293T cells were seeded in 6-well plates at 30% confluence the day before transfection. The total amount of DNA per well was adjusted to 1 μg and included 100 ng of the tfap2α promoter:luciferase and 100 pg of pRL-CMV. All plasmids were transfected using X-tremeGENE HP as described above. 24 hr after transfection, cells were lysed and processed using the Dual-Luciferase Reporter Assay System protocol (Promega, Madison, WI). For in vivo experiments, 100 pg of the tfap2α:luciferase promoter and 10 pg of pRL-CMV were injected at the 8 cell stage in a dorsal animal blastomere. Whole embryos or dissected CNC explants were lysed and processed following the Dual-Luciferase Reporter Assay System protocol. The luminescence signals were measured using a luminometer microplate reader (BMG LABTECH Ortenberg, Germany).

## Statistical methods

For in vivo experiments, each experiment was repeated at least three times on separate days using on average 30 embryos per case per experiment. For Q-PCR and luciferase assays, all experiments were performed at least three times (biological repeats) and each PCR was performed in triplicate. Error bars represent standard error to the mean. One-way ANOVA (non-parametric) was performed using Newman–Keuls to determine statistical significance. Statistically significance was set at $*p<0.05$, $**p<0.01$ and $***p<0.005$.

## Acknowledgement

This work was supported by the National Institutes of Health, U.S. Public Health Service, Grant RO1-DE016289 and R24-OD021485 to DA, F31-DE023275 to GA and RO3-DE025692 to HC. The author would like to thank Vanessa Bartollo, Peri Prendergast, Erin Kerdavid, Siddheshwari Advani, Julie Fletcher, Shreyas Srikanth and Brittany Woodland for their contribution to experiments and discussion of this manuscript.

## Additional information

### Funding

| Funder | Grant reference number | Author |
| --- | --- | --- |
| National Institute of Dental and Craniofacial Research | F31-DE023275 | Genevieve Abbruzzese |
| National Institute of Dental and Craniofacial Research | RO3-DE025692 | Helene Cousin |
| National Institute of Dental and Craniofacial Research | RO1-DE016289 | Dominique Alfandari |
| National Institutes of Health | R24-OD021485 | Dominique Alfandari |

The funders had no role in study design, data collection and interpretation, or the decision to submit the work for publication.

## Author contributions
Vikram Khedgikar, Conceptualization, Formal analysis, Supervision, Investigation, Methodology, Writing—original draft, Writing—review and editing; Genevieve Abbruzzese, Conceptualization, Formal analysis, Methodology, Writing—review and editing; Ketan Mathavan, Formal analysis, Investigation, Methodology, Writing—review and editing; Hannah Szydlo, Validation, Investigation; Helene Cousin, Formal analysis, Supervision, Methodology, Writing—review and editing; Dominique Alfandari, Conceptualization, Formal analysis, Supervision, Funding acquisition, Methodology, Writing—review and editing

## Author ORCIDs
Dominique Alfandari http://orcid.org/0000-0002-0557-1246

## Ethics
Animal experimentation: This study was performed in strict accordance with the recommendations in the Guide for the Care and Use of Laboratory Animals of the National Institutes of Health. All of the animals were handled according to approved institutional animal care and use committee (IACUC) protocols (#2015-0029) of the University ofMassachusetts Amherst.

## Decision letter and Author response
Decision letter https://doi.org/10.7554/eLife.26898.029
Author response https://doi.org/10.7554/eLife.26898.030

# Additional files

## Supplementary files
• Transparent reporting form
DOI: https://doi.org/10.7554/eLife.26898.028

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
