## [Decision Letter]

Thank you for submitting your article "Dual control of PCNS expression and function by ADAM13/33 via AP2a and Arid3a" for consideration by *eLife*. Your article has been favorably evaluated by Marianne Bronner (Senior Editor) and three reviewers, one of whom is a member of our Board of Reviewing Editors. The following individuals involved in review of your submission have agreed to reveal their identity: Jean-Pierre Saint-Jeannet (Reviewer #2); Mike Klymkowsky (Reviewer #3).

The reviewers have discussed the reviews with one another and the Reviewing Editor has drafted this decision to help you prepare a revised submission. Given the list of essential changes, including several new experiments, the Board and reviewers invite you to respond soon with an action plan and a timetable for the completion of the additional work. We will share your comments with the reviewers and get back to you with recommendations.

In this manuscript the authors assign a role of ADAM13 in the regulation of AP2a, a key gene in neural crest patterning. the demonstrate that full-length ADAM13 can induce expression of the downstream target gene PCNS, and that this is dependent upon the induction of AP2a via an interaction with ARID3a. The authors show that ADAM13 cleaves the protocadherin PCNS and regulates its expression through activation of Ap2α, an important neural plate border specifier. Interestingly the latter activity requires both the proteolytic domain and cytoplasmic tail of ADAM13 in *Xenopus* animal cap explants. Overall this study provides important mechanistic insights into the transcriptional activity of this important class of molecules during neural crest development. While generally the results support their conclusions, there are a few areas that would strengthen their findings:

1) The necessity of ARID3a

In Figure 7, the authors show that knockdown of Arid3a, in concert with A13, can almost completely rescue the AP2a induction as well as PCNS induction. However, what is surprising is the result in Figure 7, in which AP2a/PCNS can rescue the migration defect yet Arid3a does not. This is important, since their data seems to imply an absolutely requirement for Arid3a in the actions of ADAM13. How is this explained? How was this CNC migration quantified? Representative images would be helpful and a clearer explanation of these effects is necessary. Finally, the x-axis on Figure 7 should be labelled.

2) Localization and cleavage of Arid3a by ADAM13

In the Discussion, the authors propose a model whereby Arid3a is stored at the plasma membrane, and then cleaved by ADAM13 to be released into the cytoplasm and then translocated to the nucleus. This is interesting, but the authors do not show any data to support this idea, and in addition the failure of Arid3a rescue s above calls this into question. Do the authors have any immunofluorescence data to support this idea? Also, is there direct evidence of Arid3a cleavage by ADAM13 by Western blot?

3) The nuclear activity of ADAM13

Along the same lines, it is proposed that ADAM13, either alone or with Arid3a, may act directly in the nucleus and can bind to β-importin. Yet the authors state in data not shown that expressing both of these proteins together does not result in the induction of AP2a, at least in HEK293 cells. This seems at odd with their overall basic hypothesis. Can this be explained further from a mechanistic level? As noted above, the membrane to nucleus model does not seem fully able to explain this effect without additional data.

4) The interaction of ADAM13 and Arid3a

- Arid3-flag is visible in the Figure 6 lane in the absence of ADAM13. How is that explained?

- There is a need for more accurate normalization (particularly given the very high levels Arid/Fox protein and the small level of precipitated Arid/Fox proteins.

- The authors need to demonstrate the specificity of the various anti-ADAM13 antibodies used in this analysis, at the very least by presenting an immunoblot of whole cell and whole embryo lysates probed with the antibody.

- Have the authors seen the co-localization of ADAM13 with either Arid or Fox proteins within embryos? Do they see interactions between the cytoplasmic tail of ADAM13 and either protein?

5) PCNS effects

The ADAM13 morpholino/down regulation (MO) phenotype can be partially ameliorated (rescued) by expression of either a fragment of pcdh8l and that ADAM13 can cleave pcdh8l when constructs are co-transfected into human HEK293T cells. Has this observation be repeated in *Xenopus* embryos? In Figure 2) it is difficult to tell if the relative amounts of full length and cleaved pcdh8l are the same in the control and MO treated embryos.

What is the extent of ADAM13 down-regulation in the ADAM13 MO embryos? The significance of these rescue studies is unclear since, the authors note that the protocadherin PAPC can replace pcdh8l. As noted later on in the manuscript, cadherin-11 and PAPC can rescue ADAM13 MO phenotypes; do cadherin-11 or PAPC influence pcdh8l expression or cleavage alone or in ADAM13 MO embryos? Does expression of pcdh8l alter cadherin-11 or PAPC expression levels? In terms of genetic analysis are we looking at various forms of by-pass suppressor?

In their animal cap experiments, injection of ADAM13 RNA induced increased pcdh8l expression, it appears (based on data from Xenbase) that ADAM13 is expressed maternally. ADAM13 RNA appears to be supplied maternally, does the ADAM13 MO reduced pcdh8lRNA/protein levels in animal caps? Does ADAM13 RNA also increase cadherin-11 or PAPC RNA (see above)? The authors should note the relative level of endogenous ADAM13 to exogenous ADAM13 in these studies.

6) The neural crest effects of ADAM13

It is remarkable that ADAM13 is sufficient activate Ap2α and PCNS in animal cap explants, and it would be interesting to know whether ADAM13 can induce a full program of neural crest differentiation in this context. There is little information in the manuscript on the expression pattern of Arid3a as it relates to neural crest development. The authors should comment on the timing/sequence of expression of these factors, as the model predicts that ADAM13 and Arid3a act upstream of Ap2α and PCNS.

[Editors' note: further revisions were requested prior to acceptance, as described below.]

Thank you for sending us your revised manuscript and response to reviewers. These have been considered by the handling *eLife* editors and the referees, and we are pleased to inform you that the assessment has been generally positive. However, the editors and referees have these outstanding concerns that we would like to ask you to address in a revision:

I still find that parts of the manuscript are written in a needlessly confusing way, and there are still strings of meaningless characters here and there within, at least, my copy.

First, perhaps the Abstract could start with:

"Adam13/33 is a cell surface metalloprotease critical for cranial neural crest (CNC) cell migration; it can cleave multiple substrates including itself, fibronectin, ephrinB, and cadherin-11, and pcdh8l/PCNS (this work)."

In the Abstract, it appears that the gene regulatory effects of ADAM13 are directly mediated by the proteolytic cleavage Arid3a, which in turn influences the activity of Tfap2a. This should be made clearer.

I would remove "post-translational modification" for clarity's sake.

In the Introduction, there is the impression that it is the nuclear localization of the Adam13 cytoplasmic domain that regulates gene expression; at the same time there is the observation that the interaction between Adam13 and Arid3a is critical, apparently because it is involved in a cytoplasmic complex that leads to the proteolytic cleavage of Arid3a.

Does this mean that the nuclear localization of the Adam13 cytoplasmic domain is unnecessary for Adam13's effects on gene expression?

Is there any evidence that the full length and cleaved forms of Arid3a differ in their effects on gene expression?

These points need to be clarified in the text; the reader (me) is left with the impression that Arid3a is cytoplasmic until it is proteolytically processed.

Final note; the word "this" can often be confusing, since it is not always clear what "this" refers to. Various minor typographical errors of this type should be corrected.

---

## [Author Response]

[…] While generally the results support their conclusions, there are a few areas that would strengthen their findings:1) The necessity of ARID3aIn Figure 7, the authors show that knockdown of Arid3a, in concert with A13, can almost completely rescue the AP2a induction as well as PCNS induction. However, what is surprising is the result in Figure 7, in which AP2a/PCNS can rescue the migration defect yet Arid3a does not. This is important, since their data seems to imply an absolutely requirement for Arid3a in the actions of ADAM13. How is this explained? How was this CNC migration quantified? Representative images would be helpful and a clearer explanation of these effects is necessary. Finally, the x-axis on Figure 7 should be labelled.

Figure 7 show that ADAM13 induction of AP2α or PCNS depends on the presence of Arid3a. Figure 7 shows that the loss of ADAM13, which inhibit CNC migration, can be rescued by both Ap2α or PCNS (which are both downstream of ADAM13, but not by Arid3a. This can be explained by the fact that Arid3a expression does not depends on ADAM13 (Figure 7—figure supplement 1) and that they function together. Simply put, an excess of Arid3a cannot alleviate the lack of ADAM13. We will correct the missing x-axis and will provide representative examples. Migration was scored as described in the methods by injecting a lineage tracer together with either MO and/or mRNA at the 8-cell-stage in a dorsal animal blastomere and scoring for the presence or absence of fluorescent neural crest within the migration pathway. This technique has been validated in multiple publications and compared to both ISH and grafting techniques.

2) Localization and cleavage of Arid3a by ADAM13In the Discussion, the authors propose a model whereby Arid3a is stored at the plasma membrane, and then cleaved by ADAM13 to be released into the cytoplasm and then translocated to the nucleus. This is interesting, but the authors do not show any data to support this idea, and in addition the failure of Arid3a rescue s above calls this into question. Do the authors have any immunofluorescence data to support this idea? Also, is there direct evidence of Arid3a cleavage by ADAM13 by Western blot?

It seems that our model and explanation were not sufficiently clear. We will make sure that the revised manuscript addresses the issue.

1) Arid3a is not likely to be cleaved by ADAM13 since the proteolytic domain of ADAM13 is located on the outside of the cells and Arid3a is a cytoplasmic/nuclear protein. At the moment we have no evidence that ADAM13 can cleave substrate in the cytoplasm or nucleus.

2) Arid3a has been shown to localize to lipid raft in the plasma membrane of B-cells upon Myristoylation. This localization is critical for its ability to respond to the B-cell receptor signal. Upon signaling by the BCR, arid3a translocates into the nucleus where it regulates gene expression. Our model suggests that ADAM13 may play a similar role by which it regulates Arid3a translocation to the nucleus.

We have performed multiple new experiments which are shown in Figure 10. In Hek293T cells ADAM13 increases a fragment of Arid3a in the nucleus of transfected cells (Arid3a short). This depends both on the cytoplasmic domain and proteolytic activity similar to the requirement for ADAM13 regulation of AP2a expression. We have included this new figure in the manuscript as it shows 1) that ADAM13 can regulate Arid3a post translational modification (cleavage/stability) and subcellular localization.

3) The nuclear activity of ADAM13Along the same lines, it is proposed that ADAM13, either alone or with Arid3a, may act directly in the nucleus and can bind to β-importin.

We have already demonstrated that ADAM13 cytoplasmic domain translocates into the nucleus and regulate gene expression (Cousin et al., 2011 Dev.Cell).

Yet the authors state in data not shown that expressing both of these proteins together does not result in the induction of AP2a, at least in HEK293 cells. This seems at odd with their overall basic hypothesis. Can this be explained further from a mechanistic level? As noted above, the membrane to nucleus model does not seem fully able to explain this effect without additional data.

The absence of increased transcription by the isolated cytoplasmic domain suggest that ADAM13 and Arid3a interact at the plasma membrane a location where GFP-Cyt13 is never found. We have performed membrane fractionation confirming that Arid3a is indeed associated with the plasma membrane. In Bcell, Arid3a is palmytoylated locally to the plasma membrane where it interacts with the B-cell receptor. Stimulation of the BCell receptor induce a sumoylation of Arid3a which can then go to stimulate gene expression. This is consistent with our new data and the model.

4) The interaction of ADAM13 and Arid3a- Arid3-flag is visible in the Figure 6 lane in the absence of ADAM13. How is that explained?

Hek293T cells express the human orthologue of ADAM13, ADAM33. The polyclonal used for this co-IP was raised against the DC domain of ADAM13, one of the domains that is relatively well conserved amongst ADAM. It is likely that this band represents the association with endogenous human ADAM13/33. The human protein would not be detected by the monoclonal antibody used to reprobe the IP for ADAM13; mAb7C9.

We have repeated the co-IP with another affinity purified ADAM13 antibody g821 (Author response image 1). They do confirm the interaction. We have also performed the same co-IP in the embryo and confirm the result. We propose to put the second IP (In embryos as a supplemental figure for Figure 6 or replace the 293T experiment with the embryo experiment.

- There is a need for more accurate normalization (particularly given the very high levels Arid/Fox protein and the small level of precipitated Arid/Fox proteins.

We estimate that about 1.2% of Arid3a is co-precipitated by ADAM13. Increasing the level of ADAM13 increase this amount suggesting that ADAM13 availability is the limiting factor.

For FoxD3 overexpressed in *Xenopus* embryos, about 0.4% of FoxD3 is associated with endogenous ADAM13. Again this is likely to be limited by the amount of available ADAM13.

In Hek293T cells when both proteins are co-transfected, approximately 2.3% of FoxD3 co-precipitates with ADAM13.

The co-IP demonstrates that the proteins can interact directly or indirectly. The functional evidence is the more important (Author response image 1).

In overexpression experiments, I am not sure that the proportion of protein binding is relevant.

- The authors need to demonstrate the specificity of the various anti-ADAM13 antibodies used in this analysis, at the very least by presenting an immunoblot of whole cell and whole embryo lysates probed with the antibody.

The specificity of all of the ADAM13 antibodies has been demonstrated in previous publications, but we are happy to provide those western blots.

Figure 2 IP with a goat antibody affinity purified with a fusion protein corresponding to amino acid 821 to 914 (Cousin et al., 2011, Abbruzzese et al., 2015) of ADAM13 and blotted with a rabbit polyclonal antibody to the ADAM13 cytoplasmic domain (6615F, Alfandari et al., 1997). Left lane non-injected embryos right lane embryos injected with the ADAM13 morpholino.

The same immunoprecipitating antibody was used in Figure 6 (FoxD3 co-IP) and the new embryo co-IP for Arid3a.

The same polyclonal (6615F) was used for the direct blot in Figure 6 on transfected ADAM13.

Author response image 2 shows another co-IP between ADAM13 and Arid3a with the goat affinity purified antibody directed against the cytoplasmic domain of ADAM13. This is the endogenous ADAM13.

**Author response image 2. respfig2:** 

To the left is a western blot from Hek293T transfected with ADAM13 or mouse ADAM33 or the empty vector CS2. mAb 7C9 recognize the extracellular domain of ADAM13 (Gaultier et al., 2002), rDC13 is a rabbit polyclonal to the same domain of ADAM13. G821 is a goat polyclonal antibody affinity purified with a fusion protein corresponding to the C-terminal end of the cytoplasmic domain of ADAM13 (Cousin et al., 2011, Abbruzzese et al., 2015).

- Have the authors seen the co-localization of ADAM13 with either Arid or Fox proteins within embryos?

There is no antibody to either Arid3a or FoxD3 that cross react with *Xenopus* so we cannot do this with endogenous proteins. The co-IP for FoxD3 shows that it binds to endogenous ADAM13 since this interaction is absent when ADAM13 is Knocked Down (Figure 6). The new co-IP of Arid3a from embryos also shows the same fact.

Do they see interactions between the cytoplasmic tail of ADAM13 and either protein?

We do see interaction of FoxD3 with the cytoplasmic domain of ADAM13 (see Author response image 3).

**Author response image 3. respfig3:** 

We do not see this interaction with Arid3a suggesting that the interaction may be indirect or that it requires co-expression at the plasma membrane.

5) PCNS effectsThe ADAM13 morpholino/down regulation (MO) phenotype can be partially ameliorated (rescued) by expression of either a fragment of pcdh8l and that ADAM13 can cleave pcdh8l when constructs are co-transfected into human HEK293T cells. Has this observation be repeated in Xenopus embryos?

We do not have an antibody to the N-terminus of PCNS, which mean we can only detect C-term fragments but not what is shed. We have performed mass spec experiments from either non-injected CNC or CNC injected with the ADAM13 MO. We found 5 spectra corresponding to the PCNS extracellular domain (single unique peptide) in the control CNC sup and none in the MO13 CNC sup. While it is suggestive (100% probability for this peptide), we usually want at least 2 unique peptides for identification. This is the reason we did not include this in this manuscript (Author response image 4).

**Author response image 4. respfig4:** 

We have performed overexpression of ADAM13 to test if the C-terminal fragment of PCNS increases. See new blot to the left. We propose to add this western blot as Figure 1—figure supplement 2. This blot also confirm the specificity of the 6615F antibody used to detect ADAM13.

In Figure 2) it is difficult to tell if the relative amounts of full length and cleaved pcdh8l are the same in the control and MO treated embryos.

I suppose the referee means Figure 2, since 2B is an ISH. It is difficult to evaluate the proportion of the fragment since it is barely detectable in the morphant (the signal is not significantly higher than the background in the same area). This is why we did the overexpression experiment. We believe this fragment is likely unstable explaining why it does not accumulate.

What is the extent of ADAM13 down-regulation in the ADAM13 MO embryos?

Figure 2C (A13 P and M; pro and mature form). We have been testing ADAM13 KD in each set of experiment for over a decade. It varies between 80% reduction at 18C and >90% reduction at 14C. Figure 2 shows the typical result at 14C.

The significance of these rescue studies is unclear since, the authors note that the protocadherin PAPC can replace pcdh8l. As noted later on in the manuscript, cadherin-11 and PAPC can rescue ADAM13 MO phenotypes.

The fragment of Cadherin-11, PAPC and PCNS can all partially rescue CNC migration. All of these proteins are substrate of ADAM13. We have just submitted a manuscript Methavan et al., to PlosOne that shows that the extracellular fragment of Cadherin-11 bind to growth factor receptor and modulate Akt signaling. This was also shown for the E-cadherin fragment suggesting that multiple cadherin extracellular domains may bind and signal through RTK. We will include this in the Discussion and can attach the submitted manuscript to this letter.

Do cadherin-11 or PAPC influence pcdh8l expression or cleavage alone or in ADAM13 MO embryos? Does expression of pcdh8l alter cadherin-11 or PAPC expression levels? In terms of genetic analysis are we looking at various forms of by-pass suppressor?

This question is unclear. The manuscript does not suggest that PCNS influence transcription in any way. Nor do we suggest that the extracellular fragment does.

We show that ADAM13 regulates PCNS expression. We have previously shown that ADAM13 does not regulate Cadherin-11. We also have evidence that ADAM13 regulates PAPC (microarray data published in Cousin et al., 2011). As substrate it is likely that they may compete with each other for the available ADAM13 but it is unclear how this will add to the current manuscript.

In their animal cap experiments, injection of ADAM13 RNA induced increased pcdh8l expression, it appears (based on data from Xenbase) that ADAM13 is expressed maternally. ADAM13 RNA appears to be supplied maternally, does the ADAM13 MO reduced pcdh8l RNA/protein levels in animal caps?

PCNS is barely detectable in animal cap in the absence of overexpressed ADAM13 mRNA (CT between 35 and 42). The level of maternal ADAM13 mRNA and protein is extremely low (detectable only with >PCR 35 cycles). Using northern blot ADAM13 is detected at stage 12, using RNAse protection ADAM13 is detected at stage 10. By western blot using 10 embryos, the ADAM13 protein levels is only detected at stage 10. By ISH, ADAM13 get cleared from the naïve ectoderm toward the end of gastrulation. While it is possible that the MO13 would have an effect in the animal cap, we have shown that it does in CNC which are the cells that are relevant to this study. We used the AC for gain of function because they do not express significant level of ADAM13.

While the original microarray expression data from Xenbase suggested that the level was relatively constant. The new RNAseq data (See Figure 2) shows the same results as our RNAse protection:

Does ADAM13 RNA also increase cadherin-11 or PAPC RNA (see above)? The authors should note the relative level of endogenous ADAM13 to exogenous ADAM13 in these studies.

No, ADAM13 does not induce Cadherin-11. We did not test PAPC since it is not relevant to the current study. In this study depending on the stage at which animal cap are extracted the relative level of overexpressed ADAM13 mRNA is variable. At stage 20-22, We found that the level of endogenous ADAM13 was 1/10 of what is found in a sibling embryo (Normalized to GAPDH). In the same experiment, the level of injected mRNA remaining at that stage was 100 fold compare to the non-injected animal cap and 10 fold compare to the sibling embryo.

Figure 1—figure supplement 2 shows the relative level of endogenous ADAM13 protein compare to the overexpressed protein in intact embryos.

6) The neural crest effects of ADAM13It is remarkable that ADAM13 is sufficient activate Ap2α and PCNS in animal cap explants, and it would be interesting to know whether ADAM13 can induce a full program of neural crest differentiation in this context.

We have performed new realtime qPCR with Slug, Sox8, Sox10, Twist1 and Cadherin11. ADAM13 expression robustly induces the expression of Slug and Twist but not Sox8, Sox10 or Cadherin-11. In the embryo we have shown that KD of ADAM13 does not affect Slug, Sox8 and Sox10 (Cousin et al., 2011). This gain of function experiment shows that ADAM13 can induce Slug, the loss of function shows it is not critical. We have KD both ADAM13L and S (MDC13) and in this case slug decreases. We would need to perform RNAseq to determine to what extend ADAM13 induces CNC but this is outside of the scope of this manuscript. The induction is extremely sensitive to stage. At later stage (Stage 22) we found that all of the CNC markers were no longer expressed. This argues that ADAM13 is insufficient to induce CNC.

There is little information in the manuscript on the expression pattern of Arid3a as it relates to neural crest development. The authors should comment on the timing/sequence of expression of these factors, as the model predicts that ADAM13 and Arid3a act upstream of Ap2α and PCNS.

Arid3a, ADAM13 and AP2a are all widely expressed in the ectoderm during gastrulation. All 3 genes are down regulated in the neural plate, but maintained in the neural fold. Based on the published expression pattern the mRNA and proteins are likely to be co-expressed in the neural fold during CNC specification. We are currently performing additional ISH for Arid3a but it is unclear if these will be better than those published already.

[Editors' note: further revisions were requested prior to acceptance, as described below.]

Thank you for sending us your revised manuscript and response to reviewers. These have been considered by the handling eLife editors and the referees, and we are pleased to inform you that the assessment has been generally positive. However, the editors and referees have these outstanding concerns that we would like to ask you to address in a revision:I still find that parts of the manuscript are written in a needlessly confusing way, and there are still strings of meaningless characters here and there within, at least, my copy.We have been working on the text to make it clearer, the new data have changed the story significantly but we had tried to maintain the text close to the original manuscript. We hope the new version is clearer. The string of character appears only during the pdf conversion and appears to be due to some incompatibility between Word from PC and MAC. We will make sure that we find all of the issues.First, perhaps the Abstract could start with:"Adam13/33 is a cell surface metalloprotease critical for cranial neural crest (CNC) cell migration; it can cleave multiple substrates including itself, fibronectin, ephrinB, and cadherin-11, and pcdh8l/PCNS (this work)."Done.In the Abstract, it appears that the gene regulatory effects of ADAM13 are directly mediated by the proteolytic cleavage Arid3a, which in turn influences the activity of Tfap2a. This should be made clearer.This is done. I wanted to be very cautious that we have not shown proteolysis. It could also be the stabilization of a proteolytic fragment that is independent of adam13.I would remove "post-translational modification" for clarity's sake.Done.In the Introduction, there is the impression that it is the nuclear localization of the Adam13 cytoplasmic domain that regulates gene expression.

This was the original hypothesis and is still a possibility.

At the same time there is the observation that the interaction between Adam13 and Arid3a is critical, apparently because it is involved in a cytoplasmic complex that leads to the proteolytic cleavage of Arid3a.

The complex is most likely at the plasma membrane (on the cytoplasmic side) rather than in the cytoplasm.

Does this mean that the nuclear localization of the Adam13 cytoplasmic domain is unnecessary for Adam13's effects on gene expression?Our previous work showed that introducing a single leucine in the cytoplasmic domain of adam13 that created a nuclear export signal was sufficient to prevent adam13 ability to rescue CNC migration (Cousin et al., 2011 DevCell). This strongly suggests that the cytoplasmic domain of adam13 needs to translocate to regulate gene expression. Our current results show that translocation is not sufficient. It has to go to the plasma membrane to control arid3a. It is likely that the complex (Arid3a/adam13cyto) translocates but it is not demonstrated here.Is there any evidence that the full length and cleaved forms of Arid3a differ in their effects on gene expression?

No there is no evidence that the cleaved and full length differ in their ability to regulate gene expression. These form of Arid3a have never been described before. Given the likely cleavage site the ARID domain would be maintained together with the NLS and the sumoylation site. As such it is likely that the truncated form will also regulate transcription. The N-term portion of the protein is thought to promote protein-protein interaction. We plan to test this in the future but feel that the manuscript is already long and complex without this study.

These points need to be clarified in the text; the reader (me) is left with the impression that Arid3a is cytoplasmic until it is proteolytically processed.

We will clarify.

1) There is clearly abundant full length Arid3a at the membrane in the cytoplasm and in the nucleus so proteolytic processing is not required for nuclear translocation.

2) The interaction of Arid3a with adam13 increases this proteolytic processing and the fraction of that short from present in the nucleus.

3) As described in B-cells for Bright, only a fraction of the protein is associated with lipid raft and able to respond to BCR (Here adam13) signaling.Final note; the word "this" can often be confusing, since it is not always clear what "this" refers to. Various minor typographical errors of this type should be corrected.

We will try to correct all of those before the next submission.